# Women's autonomy in healthcare decision-making and healthcare seeking behaviour for childhood illness in Ghana: Analysis of data from the 2014 Ghana Demographic and Health Survey

Eugene Budu[1], Abdul-Aziz Seidu[1,2‡]*, Ebenezer Kwesi Armah-Ansah[1], Francis Sambah[3], Linus Baatiema[1‡], Bright Opoku Ahinkorah[4‡]

**1** Department of Population and Health, University of Cape Coast, Cape Coast, Ghana, **2** College of Public Health, Medical and Veterinary Sciences, James Cook University, Townsville, Queensland, Australia, **3** Department of Health, Physical Education and Recreation, University of Cape Coast, Cape Coast, Ghana, **4** School of Public Health, Faculty of Health, University of Technology Sydney, Sydney, Australia

☯ These authors contributed equally to this work.
‡ These authors also contributed equally to this work.
* abdul-aziz.seidu@stu.ucc.edu.gh

**Data Availability Statement:** The dataset can be downloaded freely from measuredhs website

## Abstract

### Introduction

The capacity of women to decide on their healthcare plays a key role in their health. In this study, we examined the association between women's healthcare decision-making capacity and their healthcare seeking behaviour for childhood illnesses in Ghana.

### Materials and methods

We used data from the 2014 Ghana Demographic and Health Survey. A total sample of 2,900 women with children less than 5 years was used for the analysis. Data were processed and analysed using STATA version 14.0. Chi-square test of independence and binary logistic regression were carried out to generate the results. Statistical significance was pegged at 95% confidence intervals (CIs). We relied on the 'Strengthening the Reporting of Observational Studies in Epidemiology' (STROBE) statement in writing the manuscript.

### Results

Out of the 2,900 women, approximately 25.7% could take healthcare decisions alone and 89.7% sought healthcare for childhood illnesses. Women who decided alone on personal healthcare had 30% reduced odds of seeking healthcare for childhood illnesses compared to those who did not decide alone [AOR = 0.70, CI = 0.51–0.97]. With age, women aged 45–49 had 69% reduced odds of seeking healthcare for childhood illnesses compared to those aged 25–29 [AOR = 0.31, CI = 0.14–0.70]. Women from the Northern and Upper West

(https://dhsprogram.com/data/dataset/Ghana_Standard-DHS_2014.cfm?flag=0).

**Funding:** The author(s) received no specific funding for this work.

**Competing interests:** The authors have declared that no competing interests exist.

regions had 72% [AOR: 0.28, CI: 0.11–0.70] and 77% [AOR: 0.23, CI: 0.09–0.58] reduced odds of seeking healthcare for childhood illnesses respectively, compared to those from the Western region.

## Conclusion

Ghanaian women with autonomy in healthcare decision-making, those who were older and those from the Northern and Upper West regions were less likely to seek healthcare for childhood illness. To reduce childhood mortalities and morbidities in Ghana, we recommend educating women such as those who take healthcare decisions alone, older women and women from deprived regions like the Northern and Upper West regions on the need to seek healthcare for childhood illnesses.

## Introduction

Childhood illness and death have become a worldwide health priority and a Sustainable Development Goal [1,2]. Target two of the Sustainable Development Goal three (SDG 3) aims at ending preventable neonatal and under-five deaths by 2030 [3]. Studies have shown that, mortality and morbidity in children under five years have seen significant progress in the last two decades from 12.7 million in 1990 to 5.9 million in 2015 [4–6]. In most countries in sub-Saharan Africa and Central and South Asia, children are almost 80% more likely to die before the age of five than children in the high income countries [1,7]. In Ghana, the under-five mortality rate is still high with a rate of 60 deaths per 1000 live births in 2014 [8]. In most sub-Saharan African countries, including Ghana, acute childhood illnesses which include acute respiratory infections, diarrhoea, malaria, and meningitis are the leading medical causes of infant and child illnesses and deaths [1]. These conditions are both preventable and treatable with the recommended point of care and treatment at the health facility [8–11].

These illnesses cause rapid and serious physiological derangement on the baby and the time taken to seek supportive management is of essence [1]. It is therefore important to improve access to skilled health professionals, and appropriate health care-seeking behaviour of mothers in order to reduce childhood illnesses and deaths [5,12]. The risk of mortality from childhood illnesses when complicated is high and it has been established that early diagnosis and prompt treatment should occur within 24 hours of the onset of these illnesses [13]. In most low-and middle-income countries, about 10% of children die as a result of infectious diseases before their fifth birthday [14]. The causes of childhood illnesses and deaths can reduce with a timely healthcare seeking behaviour and decision making capacity of women and their families [13].

Literature shows that health seeking behaviour for childhood illnesses and deaths, and women healthcare decision-making capacity have similar associated factors including socio-economic and demographic variables [15–18]. In low-and middle-income countries, women's healthcare decision-making capacity is very important for advancement in maternal and child health outcomes and women empowerment [19,20]. The societal norms, culture, religion and other socio-cultural indicators coupled with the availability of health facilities often determine the circumstances under which women would have capacity to decide on their healthcare and healthcare behaviour of their sick child [22–24]. In Ghana, studies have shown that despite men's authority in household decision-making, women are active players in the household

decision-making process [25,26]. It has also been found that decision-making autonomy is aligned with holistic wellbeing especially in the aspect of maternal and child health and that healthcare decision-making of women in Ghana plays a role in their healthcare seeking behaviour [27]

Studies have revealed that the ability of women in low-and middle-income countries to seek healthcare services is dependent on the child's health and some other factors that interact to influence their healthcare decision making capacity and health seeking behaviour for their sick children [28–31]. These factors are not limited to their socio-economic status, perception towards modern healthcare treatment, level of education, parents' literacy level, size of the family, perceived severity of the illness and previous experience of child illness and death [28,32]. The decision-making power with access to and adequate control over economic resources by women of children have a wide positive impact on health seeking behaviour for childhood illness [33,34]. In the same way, studies from sub-Saharan Africa suggest that children of women with limited decision-making capacity in health care are at higher risk of malnourishment, child mortality and negative paediatric health [21,35,36].

In Ghana, several studies have examined the link between women's decision-making capacity and the use of maternal and child healthcare services including skilled birth attendance [27] and antenatal care, delivery and postnatal care [26]. Despite the link between healthcare decision-making capacity and health care seeking for childhood illness in other studies outside Ghana and the high under-five mortality rate in Ghana attributed to childhood illnesses, it appears no study in Ghana has explored the link between women's autonomy in healthcare decision-making and health care seeking for childhood illness. In this study, we examined the association between women's autonomy in healthcare decision-making and their health care seeking for childhood illnesses in Ghana.

## Materials and methods

### Data source

Data for the study was obtained from the 2014 Ghana Demographic and Health Survey (GDHS). Specifically, data from the children's recode file was used for the study. The GDHS is a nationwide survey conducted every five years since it began in 1988. The survey gathers information on fertility, family planning, infant and child mortality, maternal and child health as well as nutrition. The GDHS is designed to provide adequate data to monitor the population and health situation in Ghana. The survey adopts a two-stage sampling design. The first stage is characterised by the selection of clusters (427) across urban and rural (211) locations from the entire nation. These made up the enumeration areas for the study. The second stage involved the selection of households from the predefined clusters, and this resulted in the selection of 12831 households. Out of the 12831 households, a total of 5,884 women with children less than 5 years were sampled. However, in this study, only childbearing women whose children had either diarrhoea/cough/fever in the last 2 weeks prior to the survey were considered and this translated into a total sample of 2,900 women. Hence, childbearing women whose children did not have diarrhoea/cough/fever in the last 2 weeks were excluded from the study.

### Study variables

**Outcome variable.** Healthcare seeking for childhood illness was the outcome variable in this study. This variable was obtained by creating a composite variable which comprised "care seeking for diarrhoea in the last 2 weeks" and "care seeking for cough/fever in the last 2 weeks". Care-seeking for diarrhoea was derived from two sets of questions under the child

immunisation, health and nutrition section of the DHS women's questionnaire. The first question was "Has child had diarrhoea in the last 2 weeks?" The answers were 'Yes' and 'No'. Mothers who answered "Yes", were asked the second question–"Did you seek advice or treatment for the diarrhoea from any source?" The answers were 'Yes' and 'No'.

Healthcare seeking for cough/fever was derived in a similar manner to diarrhoea. In the DHS, the first question posed was "Has child had an illness with a cough/fever at any time in the last 2 weeks?" The answers were 'Yes' and 'No'. Mothers who answered 'Yes', were further asked "Did you seek advice or treatment for the cough/fever from any source?" The answers were 'Yes' and 'No' [8].

A composite variable was created and coded as "1" if care was sought in at least one of the two circumstances and "0" if no care was sought in all the two circumstances [37]. This helped to generate the outcome variable (healthcare seeking for childhood illnesses).

**Independent variables.** The key independent variable for the study was women's autonomy in healthcare decision-making. This was derived from the variable "decision on personal health care". To obtain this variable, women were asked in the DHS "who usually decides on respondent's health care" with five responses. These were "respondent alone", "respondent and husband/partner", "husband/partner alone" "someone else" and "other". This was coded as "not alone = 0" and "alone = 1". A major limitation of this variable is that it does not provide the opportunity to fully interrogate the types of women and men who tend to agree or disagree on decisions, nor the sources of disagreement [38]. Apart from the key independent variable, ten women characteristics and four child's characteristics were also considered based on their availability in the dataset. The women characteristics were age [28], region [39], educational level [28,30], marital status [3,40], wealth index [30], parity [41], ethnicity [41], place of residence [42], religion [40] and occupation [42]. The child's characteristics were birth order of the child, child's weight (weight of the child at birth) [43–46], child's twin status and child's sex [47]. Parity was coded as one birth (1), two births (2), three or more births (3). Education was recoded as no formal education (0), primary (1) and secondary/higher (2). Occupation was recoded as not working (0) and working- managerial, clerical, sales, agriculture, services and manual (1). In addition, religion was recoded into Christianity (1), Islam (2), Other (3). Child's birth order was recoded into first child (1), second child or more. Child's weight was recoded into below 2.5kg (1) and 2.5kg and above (2). This weight was generated based on studies that have considered a child's weight less than 2.5kg as low birth weight and those 2.5kg and above as non-low birth weight [35–38]. Child's twin status was recoded into single birth (1) and multiple birth (2).

## Statistical analyses

Data were processed and analyzed using STATA version 14.0, employing inferential and descriptive statistics. The analysis was done in two steps. First, descriptive statistics (frequency and percentages) were used to describe the characteristics of the respondents and their association with healthcare seeking for childhood illnesses was assessed using chi-square (see Table 1). The second step involved the use of multivariable logistic regression to assess the association between the independent variables and the dependent variable. The selection of the variables into the regression model was not influenced by their statistically significant associations with the outcome variable at the bivariate analysis but on their availability in the dataset and their significant association with healthcare seeking behaviour for childhood illnesses as found in previous studies [3,40,42]. The multivariable logistic regression analysis was conducted using three models. Model I was fitted with the key independent variable (women's autonomy in healthcare decision-making) and the dependent variable (healthcare seeking for

**Table 1. Sample characteristics and women's autonomy in healthcare decision-making and healthcare-seeking for childhood illness according to sample characteristics.**

| Variables | N | % | Healthcare seeking for childhood illnesses | | χ2 (P-value) |
|---|---|---|---|---|---|
| | | | No | Yes | |
| **Women's autonomy in healthcare decision-making** | | | | | 8.97(0.003) |
| Not Alone | 2154 | 72.26 | 6.77 | 93.23 | |
| Alone | 746 | 25.74 | 10.31 | 89.69 | |
| **Age** | | | | | 13.74(0.033) |
| 15–19 | 54 | 1.85 | 11.29 | 88.71 | |
| 20–24 | 362 | 12.47 | 7.20 | 92.80 | |
| 25–29 | 783 | 27.01 | 5.52 | 94.48 | |
| 30–34 | 820 | 28.26 | 7.83 | 92.17 | |
| 35–39 | 589 | 20.31 | 8.64 | 91.36 | |
| 40–44 | 251 | 8.65 | 8.37 | 91.63 | |
| 45–49 | 42 | 1.45 | 16.67 | 83.33 | |
| **Region** | | | | | 32.99 (<0.001) |
| Western | 265 | 9.14 | 3.11 | 96.89 | |
| Central | 267 | 9.21 | 4.06 | 95.94 | |
| Greater Accra | 598 | 20.63 | 6.23 | 93.77 | |
| Volta | 229 | 7.88 | 4.35 | 95.65 | |
| Eastern | 214 | 7.36 | 9.68 | 90.32 | |
| Ashanti | 668 | 23.05 | 11.70 | 88.30 | |
| Brong Ahafo | 252 | 8.68 | 8.26 | 91.74 | |
| Northern | 189 | 6.51 | 9.84 | 90.16 | |
| Upper East | 138 | 4.76 | 6.59 | 93.41 | |
| Upper West | 80 | 2.77 | 10.39 | 89.61 | |
| **Occupation** | | | | | 0.03(0.866) |
| Not working | 497 | 17.13 | 7.37 | 92.63 | |
| Working | 2,403 | 82.87 | 7.59 | 92.41 | |
| **Ethnicity** | | | | | 0.98(0.806) |
| Akan | 1,458 | 50.29 | 6.98 | 93.02 | |
| Ga-Adangbe | 197 | 6.78 | 7.46 | 92.54 | |
| Mole-Dagbani | 501 | 17.26 | 7.89 | 92.11 | |
| Others | 745 | 25.67 | 8.07 | 91.93 | |
| **Marital status** | | | | | 0.01(0.929) |
| Married | 2,185 | 75.36 | 7.57 | 92.43 | |
| Cohabiting | 715 | 24.64 | 7.47 | 92.53 | |
| **Residence** | | | | | 2.67(0.102) |
| Urban | 1,653 | 57.01 | 8.33 | 91.67 | |
| Rural | 1,247 | 42.99 | 6.73 | 93.27 | |
| **Educational level** | | | | | 1.28(0.526) |
| No formal education | 599 | 20.66 | 7.50 | 92.50 | |
| Primary | 466 | 16.08 | 8.70 | 91.30 | |
| Secondary/higher | 1,835 | 63.26 | 7.19 | 92.81 | |
| **Wealth quintile** | | | | | 4.30(0.367) |
| Poorest | 424 | 14.64 | 8.19 | 91.81 | |
| Poorer | 414 | 14.29 | 5.42 | 94.58 | |
| Middle | 539 | 18.59 | 7.43 | 92.57 | |
| Richer | 691 | 23.82 | 8.48 | 91.52 | |

*(Continued)*

**Table 1.** (Continued)

| Variables | N | % | Healthcare seeking for childhood illnesses | | χ2 (P-value) |
|---|---|---|---|---|---|
| | | | No | Yes | |
| Richest | 831 | 28.66 | 7.73 | 92.27 | |
| **Religion** | | | | | 1.15(0.562) |
| Christianity | 2,302 | 69.02 | 7.24 | 92.76 | |
| Islam | 502 | 22.20 | 8.32 | 91.68 | |
| Other | 96 | 4.41 | 8.87 | 91.13 | |
| **Parity** | | | | | 0.63(0.729) |
| One birth | 437 | 15.07 | 6.77 | 93.23 | |
| Two births | 704 | 24.29 | 7.33 | 92.67 | |
| Three births | 1,759 | 60.64 | 7.83 | 92.17 | |
| **Child's birth order** | | | | | 0.08(0.784) |
| One | 2,184 | 75.30 | 7.63 | 92.37 | |
| Two or more | 716 | 24.70 | 7.31 | 92.69 | |
| **Child's weight** | | | | | 1.03(0.310) |
| Below 2.5kg | 263 | 9.09 | 9.09 | 90.91 | |
| 2.5kg and above | 2,637 | 90.91 | 7.39 | 92.61 | |
| **Child's Twin status** | | | | | 0.02(0.899) |
| Single birth | 2,736 | 94.35 | 7.57 | 92.43 | |
| Multiple birth | 164 | 5.65 | 7.28 | 92.72 | |
| **Child's sex** | | | | | 1.69(0.194) |
| Male | 1,534 | 52.90 | 8.15 | 91.85 | |
| Female | 1,366 | 47.10 | 6.88 | 93.12 | |

Source: Computed from 2014 GDHS

childhood illnesses). In Model II, we adjusted for possible confounders (maternal characteristics) to assess how they affect the relationship between healthcare-seeking and women's autonomy in decision-making (Table 2). In the third model (Model III), which was the complete model, we adjusted for possible confounders (all the independent variables) to assess how they affect the relationship between healthcare-seeking and women's autonomy in decision-making. The coefficients of the models were exponentiated to derive adjusted odds ratios (AORs). Statistically significant results were assessed at 95% confidence level. To check for high correlation among the explanatory variables, a test for multicollinearity was carried out using the variance inflation factor (VIF) and the results showed no evidence of high collinearity (Mean VIF = 1.41, Maximum VIF = 2.78, and Minimum VIF = 1.01). Sample weight (v005/ 1,000,000) was used to correct for over and under-sampling while the SVY command was used to address the complex survey design and generalizability of the findings. We relied on Strengthening the Reporting of Observational Studies in Epidemiology' (STROBE) statement in writing the manuscript.

## Ethics approval

The DHS reported that ethical approval was granted by the Institutional Review Board of ICF International and Ethical Review Committee of Ghana Health Service. We further obtained permission from the DHS Program for use of this data for the study.

**Table 2. Bivariate and multivariable logistic regression analysis results.**

| Variables | Model I COR 95% CI | Model II AOR 95% CI | Model III AOR 95% CI |
|---|---|---|---|
| **Women's autonomy in healthcare decision-making** | | | |
| Not Alone | 1 | 1 | 1 |
| Alone | 0.63** [0.47–0.86] | 0.70* [0.51–0.97] | 0.70* [0.51–0.97] |
| **Age** | | | |
| 15–19 | | 0.45 [0.19–1.09] | 0.45 [0.19–1.08] |
| 20–24 | | 0.73 [0.19–1.09] | 0.74 [0.45–1.23] |
| 25–29 | | 1 | 1 |
| 30–34 | | 0.64 [0.42–1.00] | 0.65 [0.42–1.00] |
| 35–39 | | 0.59* [0.37–0.95] | 0.60* [0.37–0.97] |
| 40–44 | | 0.60 [0.34–1.08] | 0.61 [0.34–1.09] |
| 45–49 | | 0.30** [0.14–0.68] | 0.31** [0.14–0.70] |
| **Region** | | | |
| Western | | 1 | 1 |
| Central | | 0.79 [0.31–2.01] | 0.80 [0.32–2.04] |
| Greater Accra | | 0.65 [0.26–1.63] | 0.65 [0.26–1.64] |
| Volta | | 0.91 [0.35–2.38] | 0.92 [0.35–2.42] |
| Eastern | | 0.35* [0.15–0.83] | 0.36* [0.15–0.85] |
| Ashanti | | 0.28** [0.13–0.62] | 0.29** [0.13–0.63] |
| Brong Ahafo | | 0.38* [0.17–0.87] | 0.39* [0.17–0.88] |
| Northern | | 0.27** [0.11–0.69] | 0.28* [0.11–0.70] |
| Upper East | | 0.42 [0.17–1.03] | 0.42 [0.17–1.03] |
| Upper West | | 0.23** [0.09–0.57] | 0.23** [0.09–0.58] |
| **Occupation** | | | |
| Not working | | 1 | 1 |
| Working | | 1.04 [0.70–1.56] | 1.05 [0.70–1.57] |
| **Ethnicity** | | | |
| Akan | | 1 | 1 |
| Ga-Adangbe | | 0.88 [0.42–1.85] | 0.88 [0.42–1.86] |
| Mole-Dagbani | | 1.02 [0.60–1.75] | 1.03 [0.60–1.76] |
| Others | | 0.73 [0.48–1.11] | 0.74 [0.49–1.13] |
| **Marital status** | | | |
| Married | | 1.11 [0.75–1.65] | 1.12 [0.75–1.65] |
| Cohabiting | | 1 | 1 |
| **Residence** | | | |
| Urban | | 0.84 [0.55–1.29] | 0.84 [0.55–1.28] |
| Rural | | 1 | 1 |
| **Educational level** | | | |
| No formal education | | 1.05 [0.69–1.61] | 1.05 [0.68–1.60] |
| Primary | | 0.74 [0.50–1.10] | 0.74 [0.50–1.11] |
| Secondary/higher | | 1 | 1 |
| **Wealth** | | | |
| Poorest | | 0.78 [0.46–1.34] | 0.78 [0.46–1.34] |
| Poorer | | 1 | 1 |
| Middle | | 0.69 [0.40–1.20] | 0.69 [0.40–1.20] |
| Richer | | 0.63 [0.3401.17] | 0.63 [0.34–1.18] |
| Richest | | 0.69 [0.34–1.38] | 0.69 [0.34–1.39] |
| **Parity** | | | |

(*Continued*)

**Table 2.** (Continued)

| Variables | Model I COR 95% CI | Model II AOR 95% CI | Model III AOR 95% CI |
|---|---|---|---|
| One birth | | 1 | 1 |
| Two births | | 0.89 [0.54–1.47] | 0.88 [0.53–1.47] |
| Three or more births | | 0.95 [0.57–1.60] | 0.92 [0.54–1.60] |
| **Religion** | | | |
| Christianity | | 1 | 1 |
| Islam | | 1.12 [0.76–1.65] | 1.12 [0.76–1.65] |
| Other | | 0.74 [0.38–1.45] | 0.73 [0.37–1.43] |
| **Child's Birth order** | | | |
| One | | | 0.98 [0.69–1.39] |
| Two or more | | | 1 |
| **Child's Weight** | | | |
| Below 2.5 kg | | | 0.81 [0.51–1.29] |
| 2.5 kg and above | | | 1 |
| **Child's Twin status** | | | |
| Single birth | | | 0.84 [0.44–1.62] |
| Multiple birth | | | 1 |
| **Child's sex** | | | |
| Male | | | 0.83 [0.63–1.10] |
| Female | | | 1 |
| N | 2,900 | 2,900 | 2,900 |
| pseudo $R^2$ | 0.005 | 0.041 | 0.043 |

Exponentiated coefficients; 95% confidence intervals in brackets, COR = Crude Odds Ratio, AOR = Adjusted Odds Ratio, 1 = Reference category,

* $p < 0.05$,

** $p < 0.01$,

*** $p < 0.001$

Model 1: Bivariate; Model 2: Adjust for maternal characteristics; Model 3: Adjust for maternal and child characteristics.

Source: Computed from 2014 GDHS

## Results

### Sample characteristics

Table 1 shows results on the distribution of healthcare seeking for childhood illness across women's autonomy in healthcare decision-making and their background characteristics. Majority of the respondents (72.3%) did not decide on their healthcare alone, 28.3% of participants were aged 30–34, 23.1% were in the Ashanti region, 83.9% were working and 50.3% were Akans. More than three-quarter (75.4%) of the participants were married, 57.0% were in urban areas, 63.3% had secondary/higher education, 28.7% were in the richest wealth quintile and 69.0% were Christians. With the child characteristics, 90.9% of the participants had children who weighed 2.5kg and above, 94.4% of the participants had single births and 52.9% of the children were male.

### Healthcare-seeking for childhood illness

The results indicate that 93.2% of women who had no independent household decision-making capacity sought for healthcare for childhood illnesses. Women aged 25–29 had the highest

prevalence of healthcare seeking for childhood illnesses (94.5%). Women from the Western region had the highest prevalence of healthcare seeking for childhood illness (96.9%). Women who were not working (92.6%), those of the Akan ethnic group (93.0%), those who were cohabiting (92.5%), rural dwellers (93.3%), women with secondary/higher level of education (92.8%), those with poorer wealth quintile (94.6%), Christians (92.8%) and those with one birth (93.2%) had the greatest proportions of healthcare seeking for childhood illnesses. Similarly, women with two or more birth order children (92.7%), those whose children weighed 2.5kg and above (92.6%), women who had multiple births (92.7%) and those who had female children (93.1%) had the highest proportions of healthcare seeking for childhood illnesses.

### Association between women's autonomy in healthcare decision-making and healthcare seeking for childhood illnesses

Table 2 presents results on the association between women's autonomy in healthcare decision-making, maternal and child characteristics and healthcare seeking for childhood illnesses. With women's autonomy in healthcare decision-making, women who decided alone on personal healthcare had 30% reduced odds of seeking healthcare for childhood illnesses compared to those who did not decide alone [AOR = 0.70, CI = 0.51–0.97]. With age, women aged 45–49 had 69% reduced odds of seeking healthcare for childhood illnesses compared to those aged 25–29 [AOR = 0.31, CI = 0.14–0.70]. Women from the Northern and Upper West regions had 72% [AOR: 0.28, CI: 0.11–0.70] and 77% [AOR: 0.23, CI: 0.09–0.58] reduced odds of seeking healthcare for childhood illnesses respectively compared to those from the Western region.

## Discussion

Evidentially, since increase in female autonomy in sub-Saharan Africa is linked to improved maternal healthcare seeking behaviour and consequent reduction in ill health of children [24,30,48,49], this study focused on the association between women's healthcare decision-making capacity and healthcare seeking for childhood illness in Ghana. We found a relatively high prevalence of healthcare seeking for childhood illnesses among women. However, only a quarter of women in this study took healthcare decisions alone. This is in line with the findings of previous studies in Ghana [27,50].

We realised that women who take healthcare decisions alone were less likely to seek healthcare for childhood illnesses, compared to those who did not decide alone. The possible reason for the finding is that in African society, men play a paramount role in determining the health needs of a woman [51]. This is because men are considered as decision makers and those who control most of the resources in marriages and hence they decide when and where women should seek health care [52]. In this regard, women are usually not given the opportunity to visit a health facility or healthcare provider alone or to make the decision to spend money on health care [53–55]. This certainly can have serious repercussions on health in particular and self-respect in general of the women and their children. Apart from men, sometimes other family members, caretakers and friends may also be involved in the healthcare decisions of women and this limits the ability of the woman to take decisions alone [54].

Women aged 35–49 and 45–49 were less likely seek healthcare for childhood illnesses compared to those aged 25–29. In support of our finding, Ajibade et al. [32,56] found significant relationship between maternal age and health seeking behaviour, where women of similar age group as ours were less likely to seek healthcare for their sick children. It is not clear why older women might have poor healthcare seeking behaviour for sick children, but one plausible reason could be that women of this age group may have inadequate support, energy and other capacity/empowerments (such as financial, educational, relationship power dynamics) to seek

healthcare. Others may be influenced by socio-cultural factors that could affect their health seeking behaviour [23,24]. Another possible reason could be that older women may have gained experience on childcare from previous children, and hence reduced their need for healthcare seeking.

Women from the Northern and Upper West regions were less likely to seek healthcare for childhood illnesses compared to those from the Western region. Consistent with our finding, Paul and Chouhan [57] in a similar study found regional differences in maternal antenatal care (ANC) visits and child health services. The reason for this finding is not far-fetched, but we infer that individual behavioural factors could be driving these regional geographical disparities in health seeking behaviour among women, especially under a standard health insurance system and ANC health policy targeted at improving women, and child health utilisation of healthcare services [58,59]. Northern Ghana (i.e. Northern, Upper West and Upper East regions) has poorer economic and health outcomes compared to other regions in the country due to geographical, historic, and socio-cultural factors that have often excluded the north from much of Ghana's economic growth [60]. We suggest further studies that could involve a mixed method approach to better understand these regional differentials in healthcare seeking behaviour.

## Strengths and limitations

The major limitation of the study is the cross-sectional nature of the survey, which makes it impossible to draw causal interpretation between the variables; at best only associations can be drawn. Additionally, these health conditions were not based on any medical diagnosis. Data was obtained from respondents' self-report and this has the tendency to be under-reported or over-reported. Furthermore, since this is a secondary data analyses, we could not include health system/service related factors that may be related to child healthcare seeking behaviour [3]. Despite these limitations, the study has been able to unearth association between women's autonomy in healthcare decision-making and healthcare seeking behaviour for childhood illness using nationally representative dataset. This makes the conclusions generalisable to women in Ghana. The relatively large sample size used in this study made it possible to employ rigorous statistical analysis that makes the findings and conclusions from this study valid.

## Conclusion

Ghanaian women with autonomy in healthcare decision-making, those who were older and from Northern and Upper West regions were less likely to seek healthcare for childhood illness. To reduce childhood mortalities and morbidities in Ghana, we recommend educating women who take healthcare decisions alone, older women and women from deprived regions such as Northern and Upper West regions on the importance of seeking healthcare for childhood illnesses. It is also imperative to conduct a qualitative study to unravel the nuances surrounding women's autonomy in healthcare decision making and healthcare seeking for childhood illnesses.

## Supporting information

**S1 Table. STROBE 2007 (v4) statement—Checklist of items that should be included in reports of *cross-sectional studies*.**
(DOCX)

## Acknowledgments

We Acknowledge MEASURE DHS for providing us with the dataset.

## Author Contributions

**Conceptualization:** Eugene Budu.

**Data curation:** Eugene Budu.

**Formal analysis:** Eugene Budu.

**Funding acquisition:** Eugene Budu, Bright Opoku Ahinkorah.

**Investigation:** Eugene Budu, Bright Opoku Ahinkorah.

**Methodology:** Eugene Budu, Bright Opoku Ahinkorah.

**Project administration:** Eugene Budu, Bright Opoku Ahinkorah.

**Resources:** Eugene Budu, Abdul-Aziz Seidu, Bright Opoku Ahinkorah.

**Software:** Eugene Budu, Bright Opoku Ahinkorah.

**Supervision:** Abdul-Aziz Seidu, Bright Opoku Ahinkorah.

**Validation:** Eugene Budu, Abdul-Aziz Seidu, Bright Opoku Ahinkorah.

**Visualization:** Eugene Budu, Abdul-Aziz Seidu, Bright Opoku Ahinkorah.

**Writing – original draft:** Eugene Budu, Abdul-Aziz Seidu, Ebenezer Kwesi Armah-Ansah, Francis Sambah, Linus Baatiema, Bright Opoku Ahinkorah.

**Writing – review & editing:** Eugene Budu, Abdul-Aziz Seidu, Ebenezer Kwesi Armah-Ansah, Francis Sambah, Linus Baatiema, Bright Opoku Ahinkorah.

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
