## [Editor Report · Decision Letter 0]

25 May 2020

PONE-D-20-07266

Women’s autonomy in healthcare decision-making and healthcare seeking behaviour for childhood illness in Ghana: Analysis of data from the 2014 Ghana Demographic and Health Survey

PLOS ONE

Dear Dr. Abdul-Aziz Seidu,

Thank you for submitting your manuscript to PLOS ONE. After careful consideration, we feel that it has merit but does not fully meet PLOS ONE’s publication criteria as it currently stands. Therefore, we invite you to submit a revised version of the manuscript that addresses the points raised during the review process.

We look forward to receiving your revised manuscript.

Kind regards,

Sharon Mary Brownie

Academic Editor

PLOS ONE

 Editor Comments:

Please clarify the following comment before your manuscript is sent out for review.

......'women whose children weighed 50-100 kg, and 101-150kg were more likely to seek healthcare for childhood illnesses'

These weight ranges are not commensurate with childhood weight averages

Journal Requirements:

2. In your Methods section, please provide additional information about the  demographic details of your participants. In paritcluar, please clarify how "child weight" was categorised, and ensure that the numbers reported are correct, as it seems unlikely that children aged 0-5 years weighted up to 200 kg.

3. Please correct your reference to "p=0.000" to "p<0.001" or as similarly appropriate, as p values cannot equal zero.
---

## [Author Response · Author response to Decision Letter 0]

31 May 2020

Abdul-Aziz Seidu

University of Cape Coast, Ghana 

Department of Population and Health 

31/05/2020

Dear Editor, 

Thank you for giving us the opportunity to revise our manuscript before sending it out for peer review. We have revised it based on the issues raised. 

1. Comment: ......'women whose children weighed 50-100 kg, and 101-150kg were more likely to seek healthcare for childhood illnesses'

Response: Please we have corrected this. We have corrected to “Child’s weight was recoded into below 2.5kg (1) and 2.5kg and above (2). This weight was generated based on studies that have considered a child’s weight less than 2.5kg as low birth weight and those 2.5kg and above as non-low birth weight [35-38].

2. Comment: These weight ranges are not commensurate with childhood weight averages

Response: Child’s weight was recoded into below 2.5kg (1) and 2.5kg and above (2). This weight was generated based on studies that have considered a child’s weight less than 2.5kg as low birth weight and those 2.5kg and above as non-low birth weight [35-38].

Comment: 3. Please ensure that your manuscript meets PLOS ONE's style requirements, including those for file naming. The PLOS ONE style templates can be found at https://journals.plos.org/plosone/s/file?id=wjVg/PLOSOne_formatting_sample_main_body.pdf and https://journals.plos.org/plosone/s/file?id=ba62/PLOSOne_formatting_sample_title_authors_affiliations.pdf

Response: Please we have formatted the manuscript to met PLOS ONE's style requirements

Comment: 4. In your Methods section, please provide additional information about the demographic details of your participants. In particular, please clarify how "child weight" was categorised, and ensure that the numbers reported are correct, as it seems unlikely that children aged 0-5 years weighted up to 200 kg.

Response: Child’s weight was recoded into below 2.5kg (1) and 2.5kg and above (2). This weight was generated based on studies that have considered a child’s weight less than 2.5kg as low birth weight and those 2.5kg and above as non-low birth weight [35-38].

Comment: 5. Please correct your reference to "p=0.000" to "p<0.001" or as similarly appropriate, as p values cannot equal zero.

Response: Please this has been corrected. Thank you. 

Yours Sincerely,

On behalf of the authors

Abdul-Aziz Seidu

---

## [Decision Letter · Decision Letter 1]

3 Sep 2020

PONE-D-20-07266R1

Women’s autonomy in healthcare decision-making and healthcare seeking behaviour for childhood illness in Ghana: Analysis of data from the 2014 Ghana Demographic and Health Survey

PLOS ONE

Dear Dr. Abdul-Aziz Seidu,

Thank you for submitting your manuscript to PLOS ONE. After careful consideration, we feel that it has merit but does not fully meet PLOS ONE’s publication criteria as it currently stands. Therefore, we invite you to submit a revised version of the manuscript that addresses the points raised during the review process.

We look forward to receiving your revised manuscript.

Kind regards,

Sharon Mary Brownie

Academic Editor

PLOS ONE

Editor Comments

Reviewers have provided some comprehensive feedback. Please consider these carefully and respond to each recommendation.

Reviewers' comments:

Reviewer's Responses to Questions

**Comments to the Author**

1. If the authors have adequately addressed your comments raised in a previous round of review and you feel that this manuscript is now acceptable for publication, you may indicate that here to bypass the “Comments to the Author” section, enter your conflict of interest statement in the “Confidential to Editor” section, and submit your "Accept" recommendation.

Reviewer #1: All comments have been addressed

Reviewer #2: All comments have been addressed

2. Is the manuscript technically sound, and do the data support the conclusions?

Reviewer #1: Partly

Reviewer #2: Yes

3. Has the statistical analysis been performed appropriately and rigorously? 

Reviewer #1: No

Reviewer #2: Yes

4. Have the authors made all data underlying the findings in their manuscript fully available?

Reviewer #1: Yes

Reviewer #2: Yes

5. Is the manuscript presented in an intelligible fashion and written in standard English?

Reviewer #1: No

Reviewer #2: Yes

6. Review Comments to the Author

Reviewer #1: Overall comment

The study assessed the relationship between women’s autonomy in healthcare decision-making and their healthcare seeking behaviour for childhood illnesses in Ghana. It highlights that healthcare seeking behaviour is good with >90% seeking care and that only 7% of Ghanaian women have autonomy in healthcare decision-making. Women autonomy could be a double-edged sword reflecting women empowerment as well as be a barrier to healthcare, as highlighted in this study where autonomous women were less likely to seek care. While the findings are key in ensuring that "no one is left behind" in Ghana, the study requires major revisions, which are outlined below.

Abstract

1. Revise the stated objective to make it specific and measurable and consistently use it throughout the manuscript (see lines 94-95)

2. Be specific about what statistical analyses were performed and why. Descriptive and inferential statistics is too general and does not point out how the research questions were answered.

3. The abstract results show provide some descriptive findings to provide context for a reader e.g. include information on sample characteristic, % of women seeking healthcare for childhood illness and % of women who are autonomous in decision making

Introduction

4. The introduction is focused on global/regional context. Local (Ghana) context is missing despite available studies exploring acute childhood illnesses, women autonomy, healthcare decision-making and health seeking behaviours including the Ghana Demographic Health Surveys

5. It is not clear from the introduction, what the research problem is. While authors note that there is absence of studies on the subject, the authors have argued why it is important for the subject to be studied and how the specific linkage, in particular, is important in Ghana.

6. Line 57 and 68: The authors refer to low- and middle-income regions. This reference is too broad and it is not clear whether there are regions that are LMI but they are countries within a region that are HIC, MIC or LIC. I suggest they use low- and middle-income countries, which is more specific.

7. The authors should correct reference their work e.g. Line 76, the authors reference a single Ethiopian studies in a general statement about developing countries; Lines 57-59, references 1 and 7 refer to WHO yet the cited information does not correctly represent the information in the references [The authors cite low- and middle-regions yet the reference talks of Sub-Saharan Africa and Central and South Asia].

Materials and Methods

8. Line 107: How many households were selected?

9. The authors should provide a clear breakdown of how the study sample was arrived at e.g. Out of the 11835 HH interviewed in the survey, how many women were interviewed and how many of those women had children ≤5 years and how many of them were married. It would also important to know the total number of children included

10. Outcome variables: The authors should clarify how they dealt with the women with children who did not have diarrhoea, fever, or cough in the last 2 weeks.

11. What was the conceptual definition of “women’s autonomy in healthcare decision-making”? How did the authors deal with joint decision making and decisions by other people e.g. caretakers etc.? The limitations of the key independent variable should be discussed e.g. Seymour and Peterman (2017), Understanding the Measurement of Women’s Autonomy Illustrations from Bangladesh and Ghana (http://ebrary.ifpri.org/utils/getfile/collection/p15738coll2/id/131367/filename/131578.pdf) discusses these limitations.

12. All the study variables should be clearly operationalised and source reference included. I suggest the author reference specific studies where each study variable was obtained from instead of the general statement “from previous studies”

13. I suggest line 139-140 be deleted and lines 137-139 be moved much earlier in the section.

14. Child weight: Are the authors referring to birth weight or the child weight during the survey? If it is the former, how is it related with childhood illness / healthcare seeking today? What was the rationale for categorisation of the continuous variable – child weight? Why did the authors categorise it into two instead of three (LBW, Normal, Overweight)?

15. It is not clear how the authors recoded the various independent variables e.g. education is recoded up to secondary education while occupation is recoded as working/not working. Were there women with tertiary (college/university) education? What does no education mean – no formal education or? For occupation, what does ‘working’ refers to? Are homemakers included? How were women on self-employment classified?

16. What informed the categorisation of age? Why was age not used as continuous variable or categorised into 10 years age-group? What informed the choice of 25-29 years as the reference category?

17. The use of the term influence throughout the manuscript may be construed to infer causation, yet these is an association study. I suggest the authors revise the manuscript (use association/relationship instead) to reflect the same.

18. The description of Model II and III is incorrect. These are multivariable logistic regressions and not binary logistic regression as stated. Also, the purpose of the models is to adjust for possible confounders and assess how they affect the relationship between healthcare-seeking and women’s autonomy in decision-making.

19. The use of variance inflation factor (VIF) in this study is not clear. All the study variables are categorical. My understanding is that VIF cannot be use with categorical variable because it is suitable with variables having 1 degree of freedom (which is not the case of categorical variable). Instead, generalized VIF can be used for categorical variables in R [not sure in Stata]

20. The method section should be referenced appropriately – Information on the data, study variables e.g. questions and ethical approvals should be referenced to the DHS

21. Could the inclusion of all the variables have affected the findings? Was the final model the best fit or the most robust model? I suggest the authors consider reviewing the inclusion of variables in the final model (https://www.ncbi.nlm.nih.gov/pmc/articles/PMC2633005/ and https://mybiostats.files.wordpress.com/2015/03/model_building_strategies_and_methods_for_logistic_regression.pdf)

22. I suggest that the authors use the STROBE reporting guidelines

Results

23. The authors should consider presenting percentages up to one decimal place. The additional decimal place does not improve precision of the findings. Also consider making the findings clear and concise.

24. Suggest that the authors organise their results using the following sub-titles: (a) Sample characteristics (b) Healthcare-seeking for childhood illness (c) Women’s autonomy in healthcare decision-making (d) Association between women’s autonomy in healthcare decision-making and healthcare seeking for childhood illnesses. The tables could be: Table 1. Sample characteristics; Table 2. Women’s autonomy in healthcare decision-making and Healthcare-seeking for childhood illness according to sample characteristics; Table 3: Binary and multivariable logistic regression analysis (Model 1: Bivariate Model 2: Adjust for maternal Model 3: Adjust for maternal and child characteristics

25. The authors should revise the interpretation of the odds ratio e.g. Women from the Northern and Upper West regions had 72% [AOR: 0.28, CI: 0.11-0.70] and 73% [AOR: 0.23, CI: 0.09-0.58] reduced odds of seeking healthcare for childhood illnesses respectively compared to those from the Western region.

Discussion

26. Line 220-222: The study findings do not support the statement “…issues of maternal utilization of these services limited by female autonomy in healthcare decision-making”. The study findings show that women autonomy is in factor a barrier to healthcare seeking behavior and that the level of healthcare seeking for childhood illness is very high in Ghana.

27. From the study findings, there is good health seeking practices for childhood illness regardless of the maternal and child characteristics. This should be discussed. The characteristics of women who were autonomous (made decisions alone) would help understand health seeking behaviours.

28. What is the implication on the study of the high proportion (72.3%) of women who did not decide on their healthcare alone? This should be discussed.

29. The study findings showed that making decision alone reduced the odds of healthcare seeking for childhood illness by 30%. The argument line 228-238 is focussed on limitation of women to make their own decision and how it affects their health-seeking behaviours. However, it does not address why the “women who made decision alone were less likely to seek healthcare for the childhood illness”. Importantly, the study findings indicate that healthcare seeking is not a problem in Ghana with >90% of women seeking care for childhood illness hence the argument that “low status of women can hinder them from recognizing and voicing their concerns about health needs even when it comes to seeking healthcare for their children” may not be valid. Also, the study does not explore the characteristics of women who make decisions alone, hence it is difficult to support these lines of discussion.

30. Line 230: Reference is made to men only yet decision making involves other people such as the extended family or caretakers

31. Line 239-247: The discussion is focussed on the 45-49 year but does not include the 35-39 years who also were less likely to seek care. The discussions should be made with reference to the women 25-29 years (are they more likely to seek care?). Is it possible that older women could also have gained experience on childcare (from previous children), hence reduced need for healthcare seeking?

Conclusion

32. I suggest the authors revise their conclusion to be in line with the study aim. E.g. Ghanaian women with autonomy in healthcare decision-making, those who were older and from Northern and Upper West regions were less likely to seek healthcare for childhood illness.

33. Revise the abstract conclusion to reflect the suggested revision in the conclusion.

34. The authors should be specific on the significance of their study, for instance, the proposed recommendation is too broad and not specific.

Major: English Editing

35. The work requires English editing to improve the clarity of some of the sentences and correct typographical and grammatical errors.

36. Revise the statement lines 55-57 for clarity. Include the specific year for the 12.7 million

37. Revise line 57-59 for clarity and to be specific. It is currently too general and unclear.

38. Lines 57-90 can be summarised into a single paragraph to reduce repetition and provide a clear and concise argument.

39. The authors should consider using low resource settings/countries instead of “developing countries”

40. Revise the statement lines 55-57 for clarity. Include the specific year for the 12.7 million

41. Line 101: Revise to reflect the correct state: The survey has so far been carried out every five-years not supposed

42. Revise “significant influence” to “association”

43. Delete line 177-178: “The data can…” This information is already stated under the data availability section.

44. Revise Line 182-183 to remove “In terms of….participants,” which is redundant.

45. Revise line 190-191 for clarity.

46. Line 215: Table 2: Revise “influence” to “Association”

47. Line 264 Include self-report

48. Revise Line 270-271 for clarity

49. Delete “In conclusion...” in line 44 and 275

Reviewer #2: The authors have addressed all comments.

However, there remains residual minor revisions which are as below:

1. The sentence on page 5 line 101, that reads “The GDHS is a nationwide survey that is supposed to be carried out every five years since it began.” The authors can state that the GDHS is a nationwide representative survey conducted every five years and if they want to add when it began they should state the year of the first survey.

2. The sentence on page 14 line 261, that reads “ This study’s major limitation is its cross-sectional which preludes causality” should be revised. It appears incomplete and hanging.

7. PLOS authors have the option to publish the peer review history of their article (what does this mean?). If published, this will include your full peer review and any attached files.

Reviewer #1: **Yes: **Samwel Gatimu

Reviewer #2: No

---

## [Author Response · Author response to Decision Letter 1]

12 Oct 2020

Abdul-Aziz Seidu

University of Cape Coast, Ghana 

Department of Population and Health 

11/10/2020

Dear Editor, 

AUTHOR’S RESPONSE TO REVIEWS:

Dear Editor, 

This letter is in reference to your email with reviewers’ comments. We are very pleased that the manuscript is potentially acceptable for publication in PLOS ONE once we have carried out some revisions. We would like to thank the reviewers for their insightful and helpful comments and for giving us the chance to revise our manuscript. We believe the revised manuscript has been significantly improved and the reviewers’ comments have been addressed adequately. We think in its current form it will make a valuable contribution to the literature on this increasingly important topic. Please find for your kind consideration the following: 1) a section-by-section response to the comments and suggestions of the reviewers (below) and 2) the revised manuscript provided as a marked-up copy and a clean copy. All changes have been marked in Yellow. We hope that these changes meet with your favourable consideration. Please do not hesitate to get in touch if you require any further information. 

Reviewer #1: Overall comment

The study assessed the relationship between women’s autonomy in healthcare decision-making and their healthcare seeking behaviour for childhood illnesses in Ghana. It highlights that healthcare seeking behaviour is good with >90% seeking care and that only 7% of Ghanaian women have autonomy in healthcare decision-making. Women autonomy could be a double-edged sword reflecting women empowerment as well as be a barrier to healthcare, as highlighted in this study where autonomous women were less likely to seek care. While the findings are key in ensuring that "no one is left behind" in Ghana, the study requires major revisions, which are outlined below.

Abstract

1. Revise the stated objective to make it specific and measurable and consistently use it throughout the manuscript (see lines 94-95)

Response: In this study, we sought to examine the association between women’s’ healthcare healthcare decision-making capacity and their healthcare seeking behaviour for childhood illnesses in Ghana. We have made this clear throughout the paper. (see line 28-30; 111-113) 

2. Be specific about what statistical analyses were performed and why. Descriptive and inferential statistics is too general and does not point out how the research questions were answered.

Response: We have made the statistical analyses more clear by stating that “Chi-square test of independence and binary logistic regression were carried out to generate the results. Statistical significance was pegged at 95% confidence intervals (CIs).” (see line 34-36)

3. The abstract results show provide some descriptive findings to provide context for a reader e.g. include information on sample characteristic, % of women seeking healthcare for childhood illness and % of women who are autonomous in decision making.

Response: We have added this “Out of the 2,450 women, approximately 25.7% could take healthcare decisions alone and 89.7% sought healthcare for their childhood illnesses (see line 39-40).

Introduction

4. The introduction is focused on global/regional context. Local (Ghana) context is missing despite available studies exploring acute childhood illnesses, women autonomy, healthcare decision-making and health seeking behaviours including the Ghana Demographic Health Surveys

Response: We have added studies in Ghana to the background (see line 66-70, 88-93). 

5. It is not clear from the introduction, what the research problem is. While authors note that there is absence of studies on the subject, the authors have not argued why it is important for the subject to be studied and how the specific linkage, in particular, is important in Ghana.

Response: We have made clear the statement of the problem by justifying why the study is important in Ghana. See line 105-111.

6. Line 57 and 68: The authors refer to low- and middle-income regions. This reference is too broad and it is not clear whether there are regions that are LMI but they are countries within a region that are HIC, MIC or LIC. I suggest they use low- and middle-income countries, which is more specific.

Response: This has been revised. We have replaced low-and middle-income regions with low-and middle-income countries (see line 83 and 94). 

7. The authors should correct reference their work e.g. Line 76, the authors reference a single Ethiopian studies in a general statement about developing countries; Lines 57-59, references 1 and 7 refer to WHO yet the cited information does not correctly represent the information in the references [The authors cite low- and middle-regions yet the reference talks of Sub-Saharan Africa and Central and South Asia].

Response: We have revised this section of the paper (see line 64-66)

Materials and Methods

8. Line 107: How many households were selected?

Response: We have given the number of households as 12831 in the methods (see line 126) 

9. The authors should provide a clear breakdown of how the study sample was arrived at e.g. Out of the 11835 HH interviewed in the survey, how many women were interviewed and how many of those women had children ≤5 years and how many of them were married. It would also important to know the total number of children included. 

Response: A breakdown of the sample has been provided in the methods (see line 124-130)

10. Outcome variables: The authors should clarify how they dealt with the women with children who did not have diarrhoea, fever, or cough in the last 2 weeks.

Response: We have made this clear by stating “In this study, only childbearing women whose children had either diarrhoea/cough/fever in the last 2 weeks were considered. Hence, childbearing women whose children did not have diarrhoea/cough/fever in the last 2 weeks were excluded from the study (see line 127-130).

11. What was the conceptual definition of “women’s autonomy in healthcare decision-making”? How did the authors deal with joint decision making and decisions by other people e.g. caretakers etc.? The limitations of the key independent variable should be discussed e.g. Seymour and Peterman (2017), Understanding the Measurement of Women’s Autonomy Illustrations from Bangladesh and Ghana (http://ebrary.ifpri.org/utils/getfile/collection/p15738coll2/id/131367/filename/131578.pdf) discusses these limitations.

Response: We have clearly indicated how women’s autonomy in healthcare decision-making was conceptualised and acknowledged the limitation of the variable (see line 151-158). 

12. All the study variables should be clearly operationalised and source reference included. I suggest the author reference specific studies where each study variable was obtained from instead of the general statement “from previous studies”

Response: We have provided specific references to all the variables associated with healthcare seeking for childhood illnesses (see line 158-163).

13. I suggest line 139-140 be deleted and lines 137-139 be moved much earlier in the section.

Response: We have deleted lines 139-140 and moved line 137-139 much earlier in the section. (see line 158-160). 

14. Child weight: Are the authors referring to birth weight or the child weight during the survey? If it is the former, how is it related with childhood illness / healthcare seeking today? What was the rationale for categorisation of the continuous variable – child weight? Why did the authors categorise it into two instead of three (LBW, Normal, Overweight)?

Response: Child weight is the weight of the child at birth. It is related because, the weight of the child at birth is associated with childhood illnesses (fever, diarhea and cough) and will therefore play a role in determining the mothers’ healthcare seeking as the child grows. The reason for the two categories (low birth weight versus non-low birth weight) was to assess how low birth weight or otherwise is linked to healthcare seeking for childhood illnesses (see line 169-170). 

15. It is not clear how the authors recoded the various independent variables e.g. education is recoded up to secondary education while occupation is recoded as working/not working. Were there women with tertiary (college/university) education? What does no education mean – no formal education or? For occupation, what does ‘working’ refers to? Are homemakers included? How were women on self-employment classified?

Response: With education, the actual coding was no formal education, primary and secondary/higher (tertiary). Higher was mistakenly left out in the methods but was included in the analysis. We have corrected this in the methods. With occupation, working refers to respondents engaged in occupations such as managerial, clerical, sales, agriculture, services and manual. These were obtained from the dataset (see line 164-166) 

16. What informed the categorisation of age? Why was age not used as continuous variable or categorised into 10 years age-group? What informed the choice of 25-29 years as the reference category?

Response: We did not recode age. In the standard demographic and health survey data that is how age is categorised and the choice of references were based on categories that had high prevalence of healthcare seeking behaviour 

17. The use of the term influence throughout the manuscript may be construed to infer causation, yet these is an association study. I suggest the authors revise the manuscript (use association/relationship instead) to reflect the same.

Response: We have revised taken out influence from appropriate aspects of the paper and replaced it with association/relationship. 

18. The description of Model II and III is incorrect. These are multivariable logistic regressions and not binary logistic regression as stated. Also, the purpose of the models is to adjust for possible confounders and assess how they affect the relationship between healthcare-seeking and women’s autonomy in decision-making.

Response: We have revised this section under statistical analysis (see line 185-191)

19. The use of variance inflation factor (VIF) in this study is not clear. All the study variables are categorical. My understanding is that VIF cannot be use with categorical variable because it is suitable with variables having 1 degree of freedom (which is not the case of categorical variable). Instead, generalized VIF can be used for categorical variables in R [not sure in Stata]. 

Response: Thanks for your comment. In STATA, VIF can be generated for categorical variables 

20. The method section should be referenced appropriately – Information on the data, study variables e.g. questions and ethical approvals should be referenced to the DHS

Response: Thank you for your comment. We have referenced information under the methods section to DHS. 

21. Could the inclusion of all the variables have affected the findings? Was the final model the best fit or the most robust model? I suggest the authors consider reviewing the inclusion of variables in the final model (https://www.ncbi.nlm.nih.gov/pmc/articles/PMC2633005/ and https://mybiostats.files.wordpress.com/2015/03/model_building_strategies_and_methods_for_logistic_regression.pdf)

Response: The variable selection was informed by variables that were significant at the bivariate level, and the model fitness was informed by the values of the pseudo R2, which was highest in the final model. 

22. I suggest that the authors use the STROBE reporting guidelines

Response: We have used this and attached it as a supplementary file (see line 197-199). 

Results

23. The authors should consider presenting percentages up to one decimal place. The additional decimal place does not improve precision of the findings. Also consider making the findings clear and concise.

Response: We have revised the results section and presented the percentages up to one decimal place (see line 208-216; 219-231)

24. Suggest that the authors organise their results using the following sub-titles: (a) Sample characteristics (b) Healthcare-seeking for childhood illness (c) Women’s autonomy in healthcare decision-making (d) Association between women’s autonomy in healthcare decision-making and healthcare seeking for childhood illnesses. The tables could be: Table 1. Sample characteristics; Table 2. Women’s autonomy in healthcare decision-making and Healthcare-seeking for childhood illness according to sample characteristics; Table 3: Binary and multivariable logistic regression analysis (Model 1: Bivariate Model 2: Adjust for maternal Model 3: Adjust for maternal and child characteristics. 

Response: We have revised these sections per your suggestion but with slight modification. For instance, instead of having two separate tables for Sample characteristics and women’s autonomy in healthcare decision-making and Healthcare-seeking for childhood illness according to sample characteristics (Table 1 and 2), we think presenting the information in one table like we already did is better. 

25. The authors should revise the interpretation of the odds ratio e.g. Women from the Northern and Upper West regions had 72% [AOR: 0.28, CI: 0.11-0.70] and 73% [AOR: 0.23, CI: 0.09-0.58] reduced odds of seeking healthcare for childhood illnesses respectively compared to those from the Western region.

Response: We have revised the interpretation of the odds ratio (see line 237-245).

Discussion

26. Line 220-222: The study findings do not support the statement “…issues of maternal utilization of these services limited by female autonomy in healthcare decision-making”. The study findings show that women autonomy is in factor a barrier to healthcare seeking behavior and that the level of healthcare seeking for childhood illness is very high in Ghana.

Response: We have taken this out of the discussion. 

27. From the study findings, there is good health seeking practices for childhood illness regardless of the maternal and child characteristics. This should be discussed. The characteristics of women who were autonomous (made decisions alone) would help understand health seeking behaviours. 

Response: We have discussed the prevalence of healthcare seeking behaviour (see line 257-266). 

28. What is the implication on the study of the high proportion (72.3%) of women who did not decide on their healthcare alone? This should be discussed. 

Response: We have discussed and given the implications of the high proportion of women who did not decide on their healthcare alone (see line 257-266). 

29. The study findings showed that making decision alone reduced the odds of healthcare seeking for childhood illness by 30%. The argument line 228-238 is focussed on limitation of women to make their own decision and how it affects their health-seeking behaviours. However, it does not address why the “women who made decision alone were less likely to seek healthcare for the childhood illness”. Importantly, the study findings indicate that healthcare seeking is not a problem in Ghana with >90% of women seeking care for childhood illness hence the argument that “low status of women can hinder them from recognizing and voicing their concerns about health needs even when it comes to seeking healthcare for their children” may not be valid. Also, the study does not explore the characteristics of women who make decisions alone, hence it is difficult to support these lines of discussion.

Response: We have taken out the invalid sentences, making the paragraph focus only on reasons why women who made decision alone were less likely to seek healthcare for the childhood illness (see line 267-277). 

30. Line 230: Reference is made to men only yet decision making involves other people such as the extended family or caretakers.

Response:: We have added a sentence to indicate that sometimes other family members, caretakers and friends are involved in decision making and this limits the ability of the woman to take decisions alone (see line 275-277).

31. Line 239-247: The discussion is focussed on the 45-49 year but does not include the 35-39 years who also were less likely to seek care. The discussions should be made with reference to the women 25-29 years (are they more likely to seek care?). Is it possible that older women could also have gained experience on childcare (from previous children), hence reduced need for healthcare seeking?

Response: We have revised this aspect of the discussion (see line 278-288).

Conclusion

32. I suggest the authors revise their conclusion to be in line with the study aim. E.g. Ghanaian women with autonomy in healthcare decision-making, those who were older and from Northern and Upper West regions were less likely to seek healthcare for childhood illness.

Response: We have revised the conclusion to read “Ghanaian women with autonomy in healthcare decision-making, those who were older and from Northern and Upper West regions were less likely to seek healthcare for childhood illness” (see line 315-316)

33. Revise the abstract conclusion to reflect the suggested revision in the conclusion.

Response: We have revised this (see line 50-55) 

34. The authors should be specific on the significance of their study, for instance, the proposed recommendation is too broad and not specific.

Response: We have revised our recommendation to read “To reduce childhood mortalities and morbidities in Ghana, we recommend educating women such as older women, women from deprived regions such as Northern and Upper West regions on the need to involve others in their healthcare decision making so as to enhance their healthcare seeking for childhood illnesses.” (see line 315-320).

Major: English Editing

35. The work requires English editing to improve the clarity of some of the sentences and correct typographical and grammatical errors.

Response: We have proof read the paper to correct the grammatical errors. 

36. Revise the statement lines 55-57 for clarity. Include the specific year for the 12.7 million

Response: We have clarified this. See line 64

37. Revise line 57-59 for clarity and to be specific. It is currently too general and unclear.

Response: We have revised this sentence. See line 64-65 

38. Lines 57-90 can be summarised into a single paragraph to reduce repetition and provide a clear and concise argument.

Response: Thank you for your suggestion. The lines 57-90 have been developed into different paragraphs that communicate different ideas. Instead of summarising into a single paragraph, we have revise the sentences to make them clear and concise. 

39. The authors should consider using low resource settings/countries instead of “developing countries”

Response: We have revised this to low and middle-income countries. See line 64 and 83.

41. Line 101: Revise to reflect the correct state: The survey has so far been carried out every five-years not supposed.

Response: We have revised this. See line 118-119

42. Revise “significant influence” to “association”

Response: We. Have revised influence to association in the entire manuscript 

43. Delete line 177-178: “The data can…” This information is already stated under the data availability section.

Response: We have deleted the information.

44. Revise Line 182-183 to remove “In terms of….participants,” which is redundant.

Response. We revised the sentence. 

45. Revise line 190-191 for clarity.

Response. We revised the sentence. See line 219-220.

46. Line 215: Table 2: Revise “influence” to “Association”

Response: We. Have revised influence to association in the entire manuscript

47. Line 264 Include self-report

Response: We have added self-report. See line 304-306.

48. Revise Line 270-271 for clarity

Response: We have revised the sentence. See line 331-332

49. Delete “In conclusion...” in line 44 and 275

Response: We have deleted the phrase. 

Reviewer #2: The authors have addressed all comments.

 However, there remains residual minor revisions which are as below:

1. The sentence on page 5 line 101, that reads “The GDHS is a nationwide survey that is supposed to be carried out every five years since it began.” The authors can state that the GDHS is a nationwide representative survey conducted every five years and if they want to add when it began they should state the year of the first survey.

Response. This has been revised. See line 118-119

2. The sentence on page 14 line 261, that reads “ This study’s major limitation is its cross-sectional which preludes causality” should be revised. It appears incomplete and hanging.

Response: We have revised the sentence. See line 302-304

Yours Sincerely,

On behalf of the authors

Abdul-Aziz Seidu

---

## [Editor Report · Decision Letter 2]

16 Oct 2020

Women’s autonomy in healthcare decision-making and healthcare seeking behaviour for childhood illness in Ghana: Analysis of data from the 2014 Ghana Demographic and Health Survey

PONE-D-20-07266R2

Dear Dr. Abdul-Aziz Seidu,

We’re pleased to inform you that your manuscript has been judged scientifically suitable for publication and will be formally accepted for publication once it meets all outstanding technical requirements.

Kind regards,

Sharon Mary Brownie

Academic Editor

PLOS ONE

 Editor Comments 
---

## [Editor Report · Acceptance letter]

26 Oct 2020

PONE-D-20-07266R2 

Women’s autonomy in healthcare decision-making and healthcare seeking behaviour for childhood illness in Ghana: Analysis of data from the 2014 Ghana Demographic and Health Survey 

Dear Dr. Seidu:

I'm pleased to inform you that your manuscript has been deemed suitable for publication in PLOS ONE. Congratulations! Your manuscript is now with our production department. 

Kind regards, 

on behalf of

Professor Sharon Mary Brownie 

Academic Editor

PLOS ONE